# Holistic processing only? The role of the right fusiform face area in radiological expertise

Ellen M. Kok[1,2]*, Bettina Sorger[3], Koos van Geel[1], Andreas Gegenfurtner[1,4], Jeroen J. G. van Merriënboer[1], Simon G. F. Robben[1,5], Anique B. H. de Bruin[1]

1 School of Health Professions Education, Maastricht University, Maastricht, The Netherlands, 2 Department of Education, Utrecht University, Utrecht, The Netherlands, 3 Department of Cognitive Neuroscience, Maastricht University, Maastricht, The Netherlands, 4 Department of Methods in Learning Research, University of Augsburg, Augsburg, Germany, 5 Department of Radiology, Maastricht University Medical Center, Maastricht, The Netherlands

* e.m.kok@uu.nl

**Data Availability Statement:** The SPSS file (without identifying information) that can be used to replicate the study's results can be accessed on Dataverse, https://doi.org/10.34894/O0CKVP.

## Abstract

Radiologists can visually detect abnormalities on radiographs within 2s, a process that resembles holistic visual processing of faces. Interestingly, there is empirical evidence using functional magnetic resonance imaging (fMRI) for the involvement of the right fusiform face area (FFA) in visual-expertise tasks such as radiological image interpretation. The speed by which stimuli (*e.g.*, faces, abnormalities) are recognized is an important characteristic of holistic processing. However, evidence for the involvement of the right FFA in holistic processing in radiology comes mostly from short or artificial tasks in which the quick, 'holistic' mode of diagnostic processing is not contrasted with the slower 'search-to-find' mode. In our fMRI study, we hypothesized that the right FFA responds selectively to the 'holistic' mode of diagnostic processing and less so to the 'search-to-find' mode. Eleven laypeople and 17 radiologists in training diagnosed 66 radiographs in 2s each (holistic mode) and subsequently checked their diagnosis in an extended (10-s) period (search-to-find mode). During data analysis, we first identified individual regions of interest (ROIs) for the right FFA using a localizer task. Then we employed ROI-based ANOVAs and obtained tentative support for the hypothesis that the right FFA shows more activation for radiologists in training versus laypeople, in particular in the holistic mode (*i.e.*, during 2s trials), and less so in the search-to-find mode (*i.e.*, during 10-s trials). No significant correlation was found between diagnostic performance (diagnostic accuracy) and brain-activation level within the right FFA for both, short-presentation and long-presentation diagnostic trials. Our results provide tentative evidence from a diagnostic-reasoning task that the FFA supports the holistic processing of visual stimuli in participants' expertise domain.

## Introduction

Radiologists have the mind-blowing ability to detect abnormalities in radiographs within 2s or even less [1]. Whereas medical students might recognize the ribs and heart but little more than

Researchers with an interest in the raw data can request access to specific files from the data manager using the contact function in Dataverse.

**Funding:** This work was supported by an fMRI scanning grant to AdB and EK from the executive board of the Faculty of Faculty of Health, Medicine and Life Sciences, Maastricht University, the Netherlands. The funding covered scanning costs for this study, but not the salary of the researchers. The funder had no role in study design, data collection and analysis, decision to publish, or preparation of the manuscript.

**Competing interests:** The authors have declared that no competing interests exist.

that, the ability to detect abnormalities develops dramatically over residency training. The question that arises is how this ability is implemented in the brain. Several studies [2–4] have investigated the neural implementation of visual expertise in radiology and other visual-expertise domains, with a focus mostly on the right fusiform face area (FFA). The exact role of the right FFA in visual-expertise domains, however, is not yet clear. In the current study, we aim to investigate the role of the right FFA by examining its involvement in the fast 'holistic mode' of diagnostic processing as compared to a slower, checking, or 'search-to-find' mode [5], using functional magnetic resonance imaging (fMRI).

## The right fusiform face area and expertise

The FFA has been found to be selectively involved in the processing of faces [6]. For example, it responds more strongly to faces than to everyday objects, and more to intact faces than to scrambled faces. Furthermore, lesions in this region have been found to cause prosopagnosia, which is the inability to recognize faces [7]. Although some researchers still argue that the FFA is uniquely dedicated to faces [8–10], there is now ample evidence to suggest that the *right* FFA plays a crucial role not only in face perception but more broadly in visual expertise [e.g., 3, 4, 11–15], as voiced in the general expertise hypothesis. Gauthier and colleagues [4] were among the first to show this effect. They found that car experts and bird experts show increased right FFA activation when looking at stimuli from their expertise domain but not from the other groups' domain. They also trained participants to recognize novel objects called greebles and found increasing activation of the right FFA with increasing expertise [16]. In participants with FFA lesions resulting in prosopagnosia, high expertise in car recognition as measured with a verbal test did not result in an equally high ability to visually recognize cars, whereas those two variables were highly correlated in healthy controls. This suggests that patients with prosopagnosia also have trouble visually individuating highly similar objects (*e.g.*, recognizing the model, manufacturer, and decade of make of cars) [17], and, likewise, that the FFA plays a role in this visual individuation of highly similar stimuli. After the classic studies of Gauthier and colleagues, expertise effects in the right FFA were established in a large number of studies with various objects of expertise [15], such as cars, [18–20], birds and minerals [21], and butterflies and moths [22]. Since it has been argued that those objects have a face or face-like structure, other investigations focused on less face-like objects such as chess boards [3,14]. Again, expertise-related activation in the right FFA was found. For example, Bilalíc and colleagues [3] found that the FFA is differently engaged in experts versus novices when chess positions were presented, but not when single chess pieces were presented [3].

Radiographs are another example of stimuli that do not resemble faces. Bilalić and colleagues found increased sensitivity of the right FFA for radiographs in experienced radiologists in comparison to medical students [23]. Similarly, Harley and colleagues found a significant correlation between diagnostic performance and right FFA-activation level in a group of expert radiologists and radiologists in training with different levels of experience [2].

Although there is evidence for the involvement of the right FFA in processing non-face visual expertise stimuli such as radiographs, it is unclear what the function of the right FFA is in these tasks. It is argued that the main role of the right FFA is in the holistic processing of faces and expertise-related stimuli [23], *i.e.*, processing a face as a whole, and not as a set of separate, distinct features that do not interact to form a single percept. Holistic processing is often evidenced by relying on the face-inversion effect: Face perception is more difficult for inverted than upright faces because inversion disrupts holistic processing [24]. Indeed, Bilalić showed an inversion effect in radiology: the right FFA of experts in radiology could distinguish upright and inverted radiographs, while the right FFA of novices could not [23].

Holistic processing is a central aspect of theories of visual expertise in general, and visual expertise in radiology in particular [25]. Visual-expertise research in radiology typically assumes a two-phase diagnostic process, consisting of a first, relatively fast, 'holistic mode' followed by a slower, 'search-to-find' or 'checking' mode [5]. The holistic mode entails an initial global analysis of the entire retinal image to distinguish normal from abnormal tissue, which subsequently guides the search to perturbations using foveal vision (the checking mode) [5]. The global impression is a comparison of the contents of the radiograph to an expert's schema of the visual appearance of normal radiographs. Central in this conceptualization of holistic processing is speed [25–27]: This global impression is developed first in the diagnostic process, and visual experts have been found to develop a global impression or gist of an image within 250–2000 ms [1] or less [5,28,29]. The slower 'checking' mode is more feature-based and involves shifting selective attention to potentially relevant areas of radiographs. Given that the right FFA is mostly linked to the holistic processing of faces and objects of expertise, it seems likely that the right FFA is less involved in the feature-based checking-mode that generally takes place after the initial holistic mode. However, investigating if this is indeed the case is only possible in experimental tasks that elicit the full diagnostic process and separates those phases. Complete separation of holistic and checking modes is not possible, but we argue that the short presentations of radiographs mostly elicits the holistic mode (since there is no time to enter the checking mode), whereas a longer presentation time in combination with the instruction to check an earlier diagnosis is expected to elicit mostly the checking mode.

Two of the neuroscientific studies that investigated radiological expertise so far aimed to capture the participants' processing of stimuli, but not specifically the diagnostic processes. They thus asked participants to execute a 1-back task [23] or a manipulation detection task [30]. In studies that did require participants to detect or diagnose abnormalities, radiographs were presented for relatively short amounts of time only, such as 500 ms [2], or 1500 ms. [31]. Tasks aimed at processing but not diagnosing radiographs, and very short tasks are likely to elicit a holistic but not a search-to-find mode. If the right FFA plays a crucial role in the process of holistic perception, it is more likely to differentiate between expertise levels if radiographs are observed for short periods of time and less so if participants engage in the slower search-to-find mode. Thus, there is a need for research that investigates the activation of the right FFA during longer presentation periods to contrast this with right FFA activation during shorter task durations, to better understand the specific function of the right FFA in the diagnostic process.

In the current study, we aim to investigate the specific function of the right FFA in visual expertise tasks in radiologists in training. To do so, we contrast laypeople with radiologists in training (residents and fellows) in a diagnostic-reasoning task. We used an established functional-localizer procedure to identify individual regions of interest (ROIs) for the right FFA, see [2,23] for following a similar approach. Localizer tasks are tasks that are known to activate a particular brain area, in this case the FFA. Of course, other brain areas are likely to be involved in visual expertise [32], [see, e.g., 30, 31 for other areas related to radiological expertise]. However, the use of a localizer task for the FFA allows us to focus our analysis on the function of the right FFA. The use of functional ROIs provides more power to detect specific differences and, to some extent, avoids the multiple-testing problem (Bennett et al. 2011). Apart from localization of the right FFA, we use similar procedures to localize the right V1 to rule out attention effects. Finally, as an exploratory analysis, we investigated the lateralization of the expertise effect. After the localizer task, we asked participants to diagnose abnormalities on radiographs after short-presentation (2s) and long-presentation (10s) times and performed ROI-based ANOVAs. Additionally, we investigated the correlation between right FFA

activation level and both diagnostic performance and radiological experience, to replicate the findings of Harley and colleagues [2]. It was hypothesized that:

1. Radiologists in training show a higher diagnostic performance than laypeople for short- and long-presentation trials.

2. For radiologists in training versus laypeople, the right FFA shows more activation during trials that elicit the holistic mode than during trials that elicit the search-to-find mode.

3. The activation level within the right FFA is positively correlated with the diagnostic performance for radiologists in training.

4. The activation level within the right FFA is positively correlated with experience level for radiologists in training.

To anticipate, we find preliminary evidence that the right FFA is selectively involved in the holistic mode.

## Materials and methods

### Participants

Participants were eleven laypeople with no experience in radiology (two males, nine females), and 17 radiologists in training: residents or fellows (seven males, ten females). An *a-priori* determined sample size of ten residents in their first year and ten residents in year 3 or higher was selected because this sample size seemed maximally feasible given the number of eligible participants at reasonable traveling distance from the MRI facilities, and was in line with earlier studies such as [2,4,16,30,31]. However, given that fewer eligible participants than expected were willing to participate, in combination with limited availability of funding, the *a priori*-determined sample size was not met. Therefore, the two groups of residents were combined into one group (radiologists in training). The average age of the laypeople was 28.4 years (*SD* = 6.2 years), nine were right-handed and two were left-handed. The average age for radiologists in training was 29.6 years (*SD* = 3.5 years), 16 were right-handed and one of them was left-handed. The experience of radiologists in training is reported in years: In the Netherlands, medical doctors specialize in radiology during a five-year residency training followed by a fellowship (one or two years) to become a subspecialist. Our sample included ten residents in their first year, one in the third year, three in their fourth year, one in the fifth year, and two first-year fellows. The laypeople were matched for age and educational level: They all had a master (*n* = 8) or PhD degree (*n* = 3) in a non-medical domain. For one participant in the laypeople condition, the behavioral data were corrupted (score is 0 due to no responses being recorded). Behavioral data from this participant were excluded from all analyses, but their fMRI data were included in all analyses. All participants had normal or corrected-to-normal vision. All participants gave written informed consent and received a compensation for study participation. This research was conducted in accordance with the Declaration of Helsinki. The Ethical Committee of the Maastricht University Medical Center approved the study protocol, file number 154066. The individual pictured in Fig 1 has provided written informed consent (as outlined in the PLOS consent form) to publish their image alongside the manuscript.

### Stimulation materials and task

**FFA-localization runs.** Individual ROIs were determined using an established FFA-localization procedure [7]: Data collection took place in two runs of 5min and 20s each with the same general structure (see Fig 1A): Images of faces (3 blocks of 30s) and objects (3 blocks of

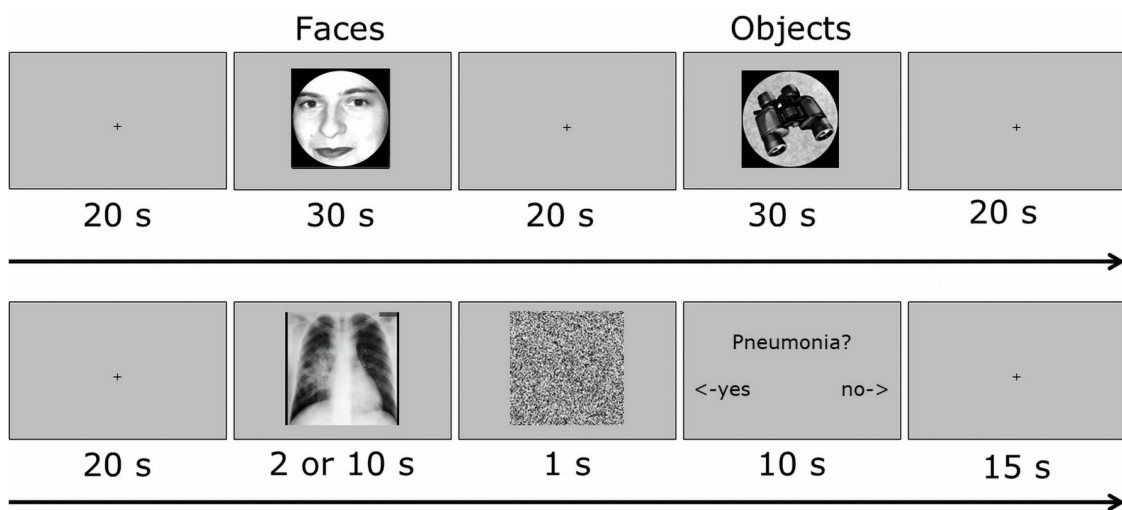

**Fig 1. Overview of the experimental designs of the FFA-localization and the diagnostic-reasoning runs.** In all runs, the stimuli were presented against a grey background. **(A)** The FFA-localization procedure used a blocked design in which 45 images of faces and 45 images of objects were shown in blocks of 30s. Between each block, there was a 20s baseline. **(B)** The diagnostic-reasoning trials consisted of the presentation of the radiograph (2s in short-presentation runs and 10s in long-presentation runs), a scrambled version of this image (mask) presented immediately after for 1s and then the diagnostic question with answer options for 10s. Between trials, there was a 15s baseline period.

30s) on a grey background were presented with a 20s resting period between each block. Blocks of faces and objects were alternating. The order of conditions was counterbalanced across the two runs. Each block consisted of 45 images that were each presented for 667ms, and participants were required to passively but attentively view the images.

**Diagnostic-reasoning runs.** For the diagnostic-reasoning runs, 66 radiographs were extracted from an existing teaching file and resized to $1000 \times 1000$ pixels with a grey background. Each of the 66 radiographs showed at least one abnormality. A total of 21 different pathologies were present (see Table 1). The diagnosis for each radiograph was extracted from the teaching file and checked by a radiologist.

The diagnostic-reasoning part of the study was split into six runs: three runs of short-presentation (2s) trials (8min and 44s per run) and three runs of long-presentation (10s) trials (11min and 40s per run), and was implemented in an event-related design. Each of the diagnostic-reasoning runs consisted of 22 radiographs. All radiographs were presented twice, first in a short-presentation run and next in a long-presentation run.

Fig 1B provides an overview of the trials. Each run started with a 20s baseline period indicated by the presentation of a black cross on a grey background. After the presentation of the radiograph, the scrambled version of the image was presented for 1s as a mask, followed by a display of the potential diagnosis (*e.g.*, "Pneumonia?") with answer options (yes/no) for 10s. For half of the radiographs, the presented diagnosis was the correct diagnosis. For the other half of the radiographs, an incorrect diagnosis was presented, which was the correct diagnosis for a randomly selected other image. The yes/no format was used because it eliminated the need for participants to speak, which can cause motion artifacts that deteriorate fMRI data quality. At the same time, by requesting participants to engage in diagnostic reasoning while the image was on the screen, we capture diagnostic reasoning processes and not just perceptual processing. Participants were requested to diagnose the image while it was on the screen, rather than afterwards (when the question was on the screen) to optimize *when* diagnostic

**Table 1. List of different abnormalities presented.**

| Name of disease/abnormality | Number of items in the experiment |
|---|---|
| Atelectasis | 4 |
| Cardiomegaly | 2 |
| COPD | 2 |
| Cystic fibrosis | 5 |
| Decompensatio cordis | 1 |
| Deviation mediastinal structures | 1 |
| Diaphragm ruptured | 1 |
| Lung fibrosis | 6 |
| Lung metastasis | 5 |
| Lung tumor | 3 |
| Lymfangitis carcinomatosa | 1 |
| Miliary tuberculosis | 5 |
| Pleural empyema | 1 |
| Pleural effusion | 6 |
| Pneumonia | 9 |
| Pneumothorax | 4 |
| Sarcoidosis | 3 |
| Silicosis | 1 |
| Broadened mediastinum | 3 |
| Enlarged hilus | 1 |
| Pleural calcification | 2 |

reasoning would take place. Participants were allowed to free-view the images, eye positions were not tracked.

Participants indicated their diagnostic decision by pressing buttons assigned to yes or no on an MRI-compatible button box. A fixation cross was subsequently presented for 15s (so the fMRI signal could return to baseline) before the next image was presented. The order of the radiographs in the three short-presentation runs was randomized for each participant, the long-presentation runs showed the radiographs in the same order as the short-presentation runs, followed by the presentation of the same potential diagnosis for the long-presentation runs as for the short-presentation runs. Participants were instructed that radiographs would be repeated in the long-presentation runs and were instructed to take another good look at the image, check their diagnosis, and if necessary adapt their answer.

## Stimulus presentation

Visual stimulation was generated by a personal computer (PC) using the *BrainStim* software (https://github.com/svengijsen/BrainStim) and projected onto a frosted screen located at the end of the scanner bore (at the side of the participant's head) with a liquid crystal display (LCD) projector. Participants viewed the screen via a mirror mounted to the head coil at an angle of ~45˚.

## (F)MRI data acquisition

Anatomical and functional brain-imaging data were obtained using a 3-T whole-body MRI scanner (Magnetom Prisma; Siemens Medical Systems, Erlangen, Germany). Participants

were placed comfortably in the MRI scanner; their heads were fixated with foam padding to minimize spontaneous or task-related motion.

**Anatomical measurements.** Each participant underwent a high-resolution T1-weighted anatomical scan using a three-dimensional (3D) magnetization-prepared rapid-acquisition-gradient-echo (MP-RAGE) sequence (192 slices, slice thickness = 1mm, no gap, repetition time [TR] = 2250ms, echo time [TE] = 2.21ms, flip angle [FA] = 9°, field of view [FOV] = $256 \times 256$mm$^2$, matrix size = $256 \times 256$, total scan time = 5min and 5s).

**Functional measurements.** Repeated single-shot echo-planar imaging (EPI) was performed using the BOLD effect as an indirect marker of local neuronal activity [33]. The number of acquisitions differed between runs (FFA-localization runs: 160 volumes; short-presentation diagnostic-reasoning runs: 262 volumes, long-presentation diagnostic-reasoning runs: 350 volumes). Apart from that, identical scanning parameters were used for all functional measurements (TR = 2000ms, TE = 30ms, FA = 77°, FOV = $192 \times 192$mm$^2$, matrix size = $96 \times 96$, number of slices = 32, slice thickness = 2mm, no gap, slice order = ascending/interleaved).

## General procedure

Before being placed in the MRI scanner, participants were informed about the study, signed informed consent, and provided information on their sex, date of birth, and year of residency. The session consisted of an anatomical scan, two FFA-localization runs, and six (three short-presentation and three long-presentation) diagnostic-reasoning runs and took 1.5-2h.

## Data analysis

Neuroimaging data were analyzed using *BrainVoyager* (v20.4, BrainInnovation BV, Maastricht, the Netherlands). Behavioral data (obtained via button presses) were extracted from the *BrainStim* logfiles and analyzed in IBM SPSS (version 22, IBM).

**Analysis of anatomical MRI data.** Anatomical images were corrected for intensity inhomogeneities and spatially normalized to Montreal Neurological Institute (MNI) space.

**Analysis of fMRI data.** Pre-processing of functional data included (a) slice-scan time correction, (b) 3D motion correction including intra-session alignment to the first functional volume of the session, (c) temporal high-pass filtering with a threshold of two cycles for the FFA-localization runs and five cycles for diagnostic-reasoning runs, and (d) spatial normalization to MNI space. Gaussian spatial smoothing (kernel: 4mm full-width at half maximum) was applied to the FFA-localization data. After 3D motion-correction was executed, the motion correction parameters were plotted and visually inspected. No runs had to be discarded for excessive, non-correctable motion.

**ROI definition.** Individual ROIs for the right and left FFA were defined by calculating individual general linear models (GLMs) with 2 (runs) × 2 predictors (face images and object images). The FFA-ROIs were defined by contrasting faces *vs.* objects. A Bonferroni-corrected statistical threshold of $p < .05$ was used. Only clusters in the right and left fusiform gyrus were included in the ROI. If clusters were too small, less stringent $p$-values were chosen until the FFA encompassed at least 20 voxels. This was necessary for nine participants (two laypeople, seven radiologists in training).

Additionally, we defined individual ROIs for the right primary visual cortex (V1). These ROIs were defined by a conjunction analysis (faces *vs.* resting and objects *vs.* resting). The most significant voxel in V1 was determined and subsequently, less stringent $p$-values were chosen until the ROI encompassed approximately 100 voxels (varying from 104 to 112 voxels).

**Investigating the effect of expertise.** Average diagnostic performance was analyzed with a 2 × 2 mixed ANOVA, with the factor *expertise level* varied between participants (laypeople and radiologists in training), and the factor *trial length* varied within participants (2s trials and 10s trials). Partial eta squared (partial $\eta^2$) was used as an effect size, where 0.02 denotes a small effect, 0.13 denotes a medium effect, and 0.26 denotes a large effect.

Four ROI-based random-effects ANOVAs were performed (for both the left FFA and the right FFA: one for long-presentation and one for short-presentation runs) that included three predictors (radiograph presentation, scrambled-radiograph presentation, and diagnosis presentation) that were separately contrasted against the baseline. The resulting four individual beta values for radiograph presentation (for both the left FFA and the right FFA: one for short-presentation and one for long-presentation runs) were extracted and further analyzed in IBM SPSS (version 22, IBM), performing first a 2 × 2 mixed ANOVAs with factors expertise level (laypeople and radiologists in training) varied between participants, trial length (2s trials and 10s trials) varied within participants and right FFA activity as the dependent variable. Next, as requested by reviewers, we also ran an exploratory 2 × 2 × 2 ANOVA with factors expertise level (laypeople and radiologists in training), trial length (2s trials and 10s trials), and location (left FFA and right FFA) and beta value as the dependent variable. Those ANOVAs were followed up by post-hoc *t*-tests with factor expertise (laypeople and radiologists in training) with Cohen's *d* as an effect size, where 0.2 denotes a small, 0.5 a medium, and 0.8 a large effect [34].

It can be argued that activity during the short trials is more reflective of holistic processing if the answer is correct. Thus, in order to investigate to what extent the hypothesized pattern was stronger when only the brain responses to correctly interpreted radiographs are analyzed (*i.e.*, those radiographs for which the participant's diagnosis was correct), an additional analysis was executed for those trials only: Consequently, four additional ROI-based random-effects ANOVAs were performed (for both the left FFA and the right FFA: one for long-presentation and one for short-presentation runs) that included four predictors (correctly diagnosed radiographs, incorrectly diagnosed radiographs, scrambled-radiograph presentation, and diagnosis presentation) that were separately contrasted against the baseline. The resulting four individual beta values for correctly diagnosed radiographs (one for short-presentation and one for long-presentation runs) were extracted and further analyzed in IBM SPSS (version 22, IBM), using the same analyses as described above.

As a check that the pattern of results was not already reflected in the early visual cortex and therewith might result from a different amount of attention paid to the stimuli (*e.g.*, experts might pay more attention to the stimuli because they are relevant for them), we repeated the analyses for hypothesis 2 in the right V1-ROI. Also, on reviewer request, we conducted ANCOVAs with V1 activity as a covariate, right FFA activity as dependent variable, and expertise (laypeople and radiologists in training) for each of the repeated measures.

Finally, the percentage of correct answers was correlated with beta values in the FFA (short-presentation and long-presentation trials separately) using the Pearson correlation coefficient. Additionally, the ordinal measure of radiological experience (in years) was correlated with the beta values in the FFA using the Spearman correlation coefficient (short-presentation and long-presentation trials separately).

## Results

### Hypothesis 1: Radiologists in training show higher diagnostic performance than laypeople for short- and long-presentation trials

Due to technical issues, the behavioral data of one participant in the laypeople group were corrupted and thus excluded here. The average diagnostic performance (percentage of correct

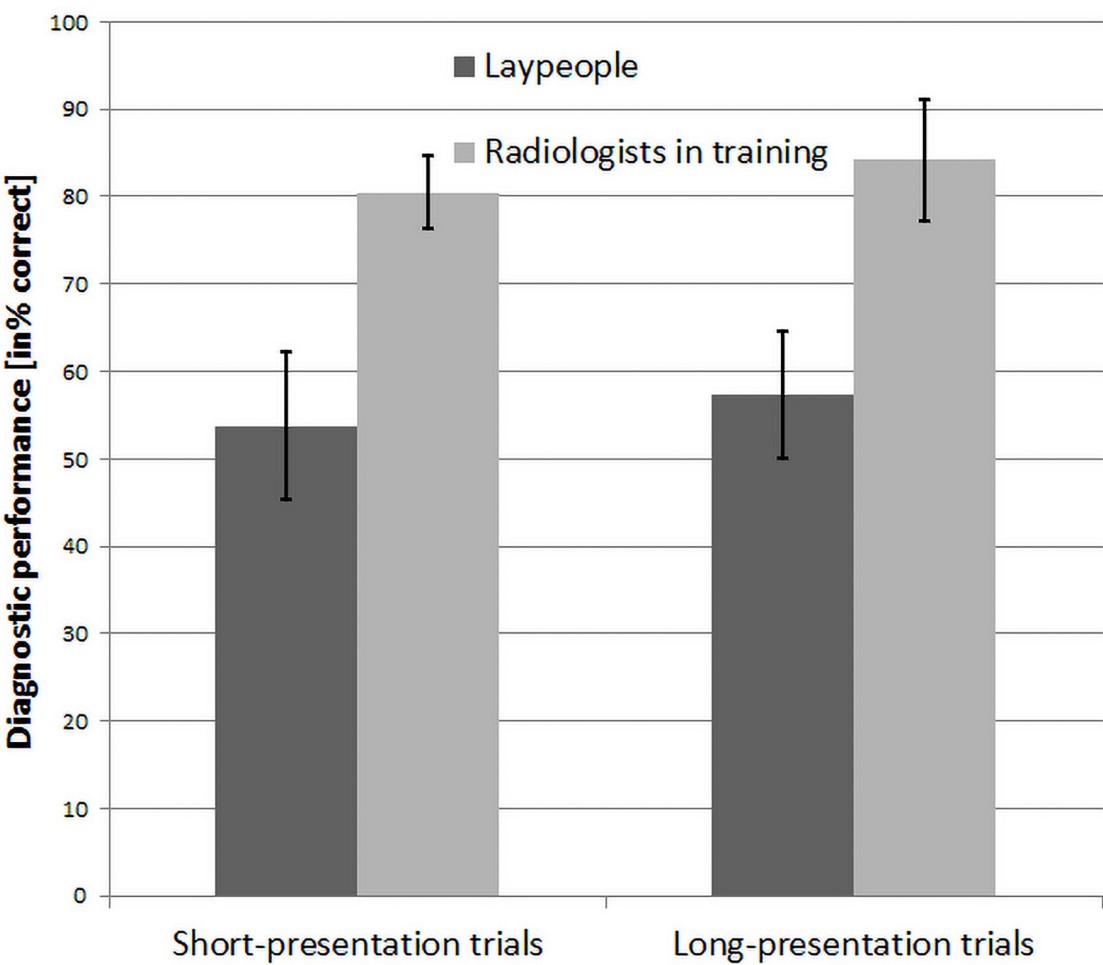

**Fig 2. Average diagnostic performance (percentage of correct diagnoses) for laypeople and radiologists in training.** Error bars show standard deviations. In both short-presentation trials and long-presentation trials, radiologists in training scored significantly higher than laypeople.

diagnoses) for short-presentation and long-presentation trials are shown in Fig 2. Laypeople scored just above chance level in both short trials ($M$ = 53.8%, $SD$ = 8.5) and long trials ($M$ = 57.3%, $SD$ = 7.3). Radiologists in training scored on average 80.5% ($SD$ = 4.1) on short trials and 84.2% ($SD$ = 7.0) on long trials. There was no interaction of *trial length* with *expertise level*, $F(1,25)$ = 0.003, $p$ = .96, $\eta^2_p < 0.001$. A main effect of *expertise level* was found, with all radiologists in training scoring higher than all laypeople, $F(1,25)$ = 142.12, $p <. 0001$, $\eta^2_p = 0.85$. Finally, there was a main effect of *trial length*, $F(1,25)$ = 6.781, $p$ = 0.02, $\eta^2_p = 0.213$, with both groups scoring higher on the long-presentation trials than on the short-presentation trials.

### Hypothesis 2: The right FFA is more activated in radiologists in training versus laypeople, during trials that elicit a holistic mode (*i.e.*, during short-presentation trials) and less so in the search-to-find mode (*i.e.*, during long-presentation trials)

A right FFA-ROI of at least 20 voxels could be defined in all participants. A probability map of the selected individual ROIs is displayed in Fig 3. The average size in voxels of the laypeople's

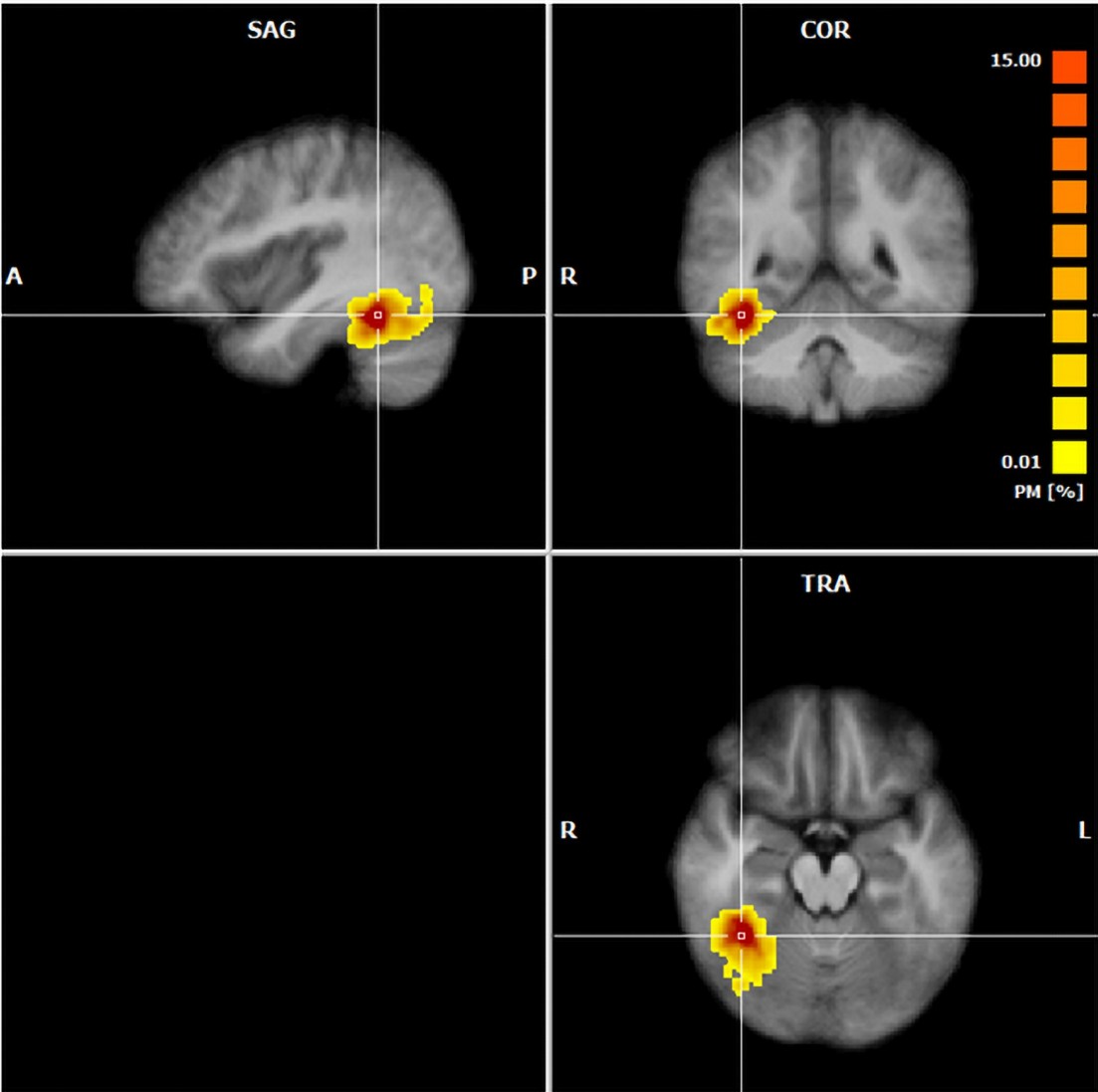

**Fig 3. Probability map, showing the locations of the individual right FFA ROIs.** Warmer colors indicate a higher proportion of ROIs in this voxel.

ROIs was 624.5 ($SD$ = 1044.2). The average size for radiologists in training was 534.5, $SD$ = 328.0.

Individual beta values within the FFA across all short-presentation and long-presentation trials separately for the two groups are depicted in Fig 4. A 2 × 2 ANOVA was run with factors *expertise level* (laypeople and radiologists in training) and *trial length* (2s trials and 10s trials) and beta value as the dependent variable. The interaction was not significant, $F(1,26) = 2.315$, $p = 0.14$, $\eta^2_p = 0.082$ (small-to-medium effect size). There was a trend towards a significant effect of trial length, $F(1,26) = 4.149$, $p = .05$, $\eta^2_p = 0.138$ (medium effect size), with lower betas for the long-presentation trials. There was no significant effect of *expertise level*, $F(1,26) = 2.152$, $p = .15$, $\eta^2_p = 0.076$ (small-to-medium effect size).

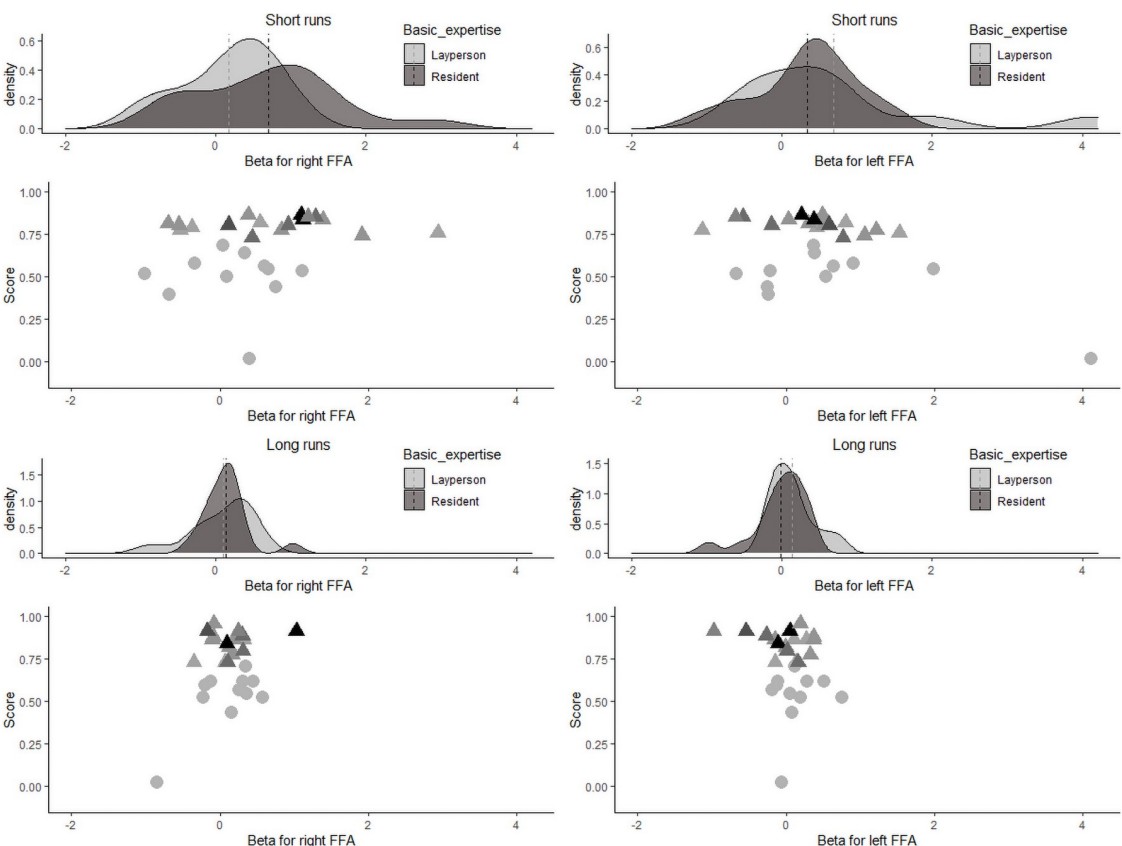

**Fig 4. Density plots (upper panels) and scatter plots (lower panels) for the individual beta values in the FFA for short- and long-presentation trials.** In the upper panels, the dotted line denotes the group mean. In the lower panels, for residents, darker colors depict more experienced residents.

We also ran an exploratory 2 × 2 × 2 ANOVA with factors *expertise level* (laypeople and radiologists in training), *trial length* (2s trials and 10s trials), and *location* (left FFA and right FFA) and beta value as the dependent variable. The three-way interaction between *expertise level*, *trial length* and *location* was significant, $F(1,26) = 4.395$, $p = .046$, $\eta^2_p = 0.145$ (medium effect size). Furthermore, the two-way interaction between *location* and *expertise level* showed a trend towards significance, $F(1,26) = 3.478$, $p = .07$, $\eta^2_p = 0.118$ (medium effect size). The two-way interactions between *trial length* and *expertise*, and between *location* and *trial length* were not significant, both $F$'s < 0.6. Finally, the main effect of *trial length* was significant, $F(1,26) = 6.388$, $p = .02$, $\eta^2_p = 0.197$ (medium-to-large effect size), but the main effects of *location* and *expertise* were not significant, both $F$'s < 0.1.

As can be seen in Fig 4, beta values were low for long trials in both the left and the right FFA for both laypeople and residents. In the short trials, beta values were higher for residents than for laypeople in the left FFA but beta values were higher for laypeople than for residents in the right FFA. Post-hoc *t*-tests show that none of those differences was significant, all *t*'s < 1.6.

**Correctly diagnosed trials only.** We additionally analyzed the mean beta value in the FFA for the correctly diagnosed trials only, see Fig 5. There was a trend towards significance for the interaction, $F(1,26) = 3.413$, $p = .08$, $\eta^2_p = .116$ (medium effect size), a main effect of

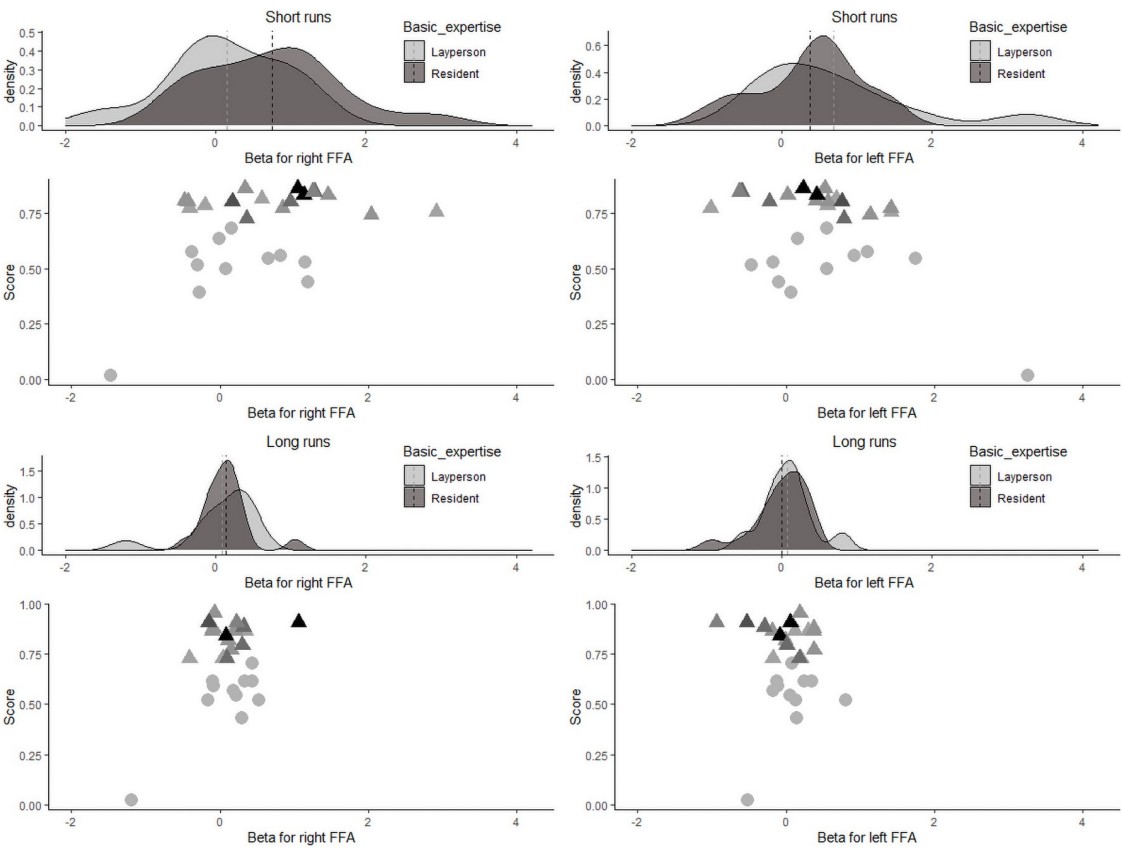

**Fig 5. Density plots (upper panels) and scatter plots (lower panels) for the individual beta values in the FFA for short- and long-presentation trials for correctly diagnosed trials only.** In the upper panels, the dotted line denotes the group mean. In the lower panels, for residents, darker colors depict more experienced residents.

*trial length*, $F(1,26) = 5.118$, $p = .03$, $\eta^2_p = 0.164$ (medium effect size), and no main effect of *expertise level*, $F(1,26) = 2.399$, $p = .13$, $\eta^2_p = .084$ (small effect size).

We also ran an exploratory $2 \times 2 \times 2$ ANOVA with factors *expertise level* (laypeople and radiologists in training), *trial length* (2s trials and 10s trials), and *location* (left FFA and right FFA) on beta value as the dependent variable. The three-way interaction between expertise level, *trial length* and *location* showed a trend towards significance, $F(1,26) = 3.937$, $p = .06$, $\eta^2_p = 0.132$ (medium effect size). There was also a trend towards significance for the interaction between *location* and *expertise level*, $F(1,26) = 2.885$, $p = .10$, $\eta^2_p = .100$ (small-to-medium sized effect). The interactions between *trial length* and *location* and *trial length* and *expertise level* were not significant, both $F$'s $< 0.7$. The main effect of trial length was significant, $F(1,26) = 10.91$, $p = .003$, $\eta^2_p = 0.296$ (large effect). The main effects of *location* and *expertise level* were not significant, both $F$'s $< 0.3$. Fig 5 shows the same pattern of results as when all trials were included. Post-hoc *t*-tests show a significant effect for the right FFA short runs, $t(26) = 1.80$, $p = .04$ (one-sided t-test), Cohen's $d = 0.71$ (medium-to-large effect); for the other *t*-tests all *t*'s $<1.0$.

**Analyses of V1 activity.** To exclude overall activation differences (already observable in the early visual cortex) between the short- and long-presentation trials, we repeated the analyses in the right V1 ROI. A $2 \times 2$ ANOVA was run with factors *expertise level* (laypeople vs.

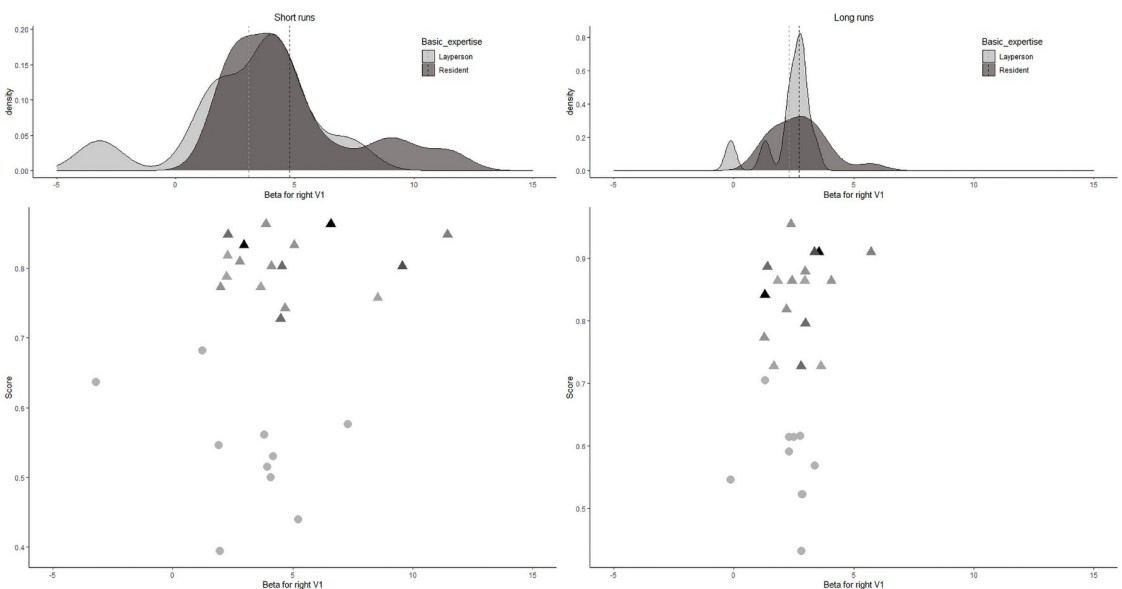

**Fig 6. Density plots (upper panels) and scatter plots (lower panels) for the individual beta values in the right V1 for short- and long-presentation trials.** In the upper panels, the dotted line denotes the group mean. In the lower panels, for residents, darker colors depict more experienced residents.

radiologists in training) and *trial length* (2s trials vs. 10s trials) and V1 activity as the dependent variable (see Fig 6).

The interaction of *expertise* with *trial length* was not significant, $F(1,26) = 1.020$, $p = 0.32$, $\eta^2_p = 0.038$ (small effect size). There was a significant effect of *trial length*, $F(1,26) = 15.821$, $p < .01$, $\eta^2_p = 0.378$ (large effect size), with lower betas for the long-presentation trials. There was no significant effect of *expertise level*, $F(1,26) = 0.581$, $p = .45$, $\eta^2_p = 0.022$ (small effect size).

We additionally analyzed the mean beta value in the V1 for the correctly diagnosed trials only (See Fig 7). There was no significant interaction, $F(1,26) = 1.614$, $p = .22$, $\eta^2_p = 0.058$ (small effect size), a main effect of *trial length*, $F(1,26) = 23.752$, $p < .01$, $\eta^2_p = 0.477$ (large effect size), and no main effect of *expertise level*, $F(1,26) = 1.128$, $p = .30$, $\eta^2_p = 0.041$ (small effect size).

Finally, as requested by reviewers, we use ANCOVAs to analyze expertise differences in short-presentation and long-presentation trials, both in all trials and only correctly diagnosed trials, with activity in V1 (during short-presentation and long-presentation trials, both in all trials or only correctly diagnosed trials) as the covariate. Table 2 shows *F*- and *p*-values for the covariate and the main effect of expertise.

### Hypothesis 3: The activation level within the right FFA is (positively) correlated with the diagnostic performance for radiologists in training

Correlations between the diagnostic performance and the right FFA activation level were calculated for radiologists in training only ($n = 17$). There was no significant correlation between the beta values for short-presentation trials and the diagnostic performance for short-presentation trials ($r = -.05$, $p = .86$), and between the beta values for long-presentation trials and the average diagnostic performance for long-presentation trials ($r = 0.24$, $p = .35$).

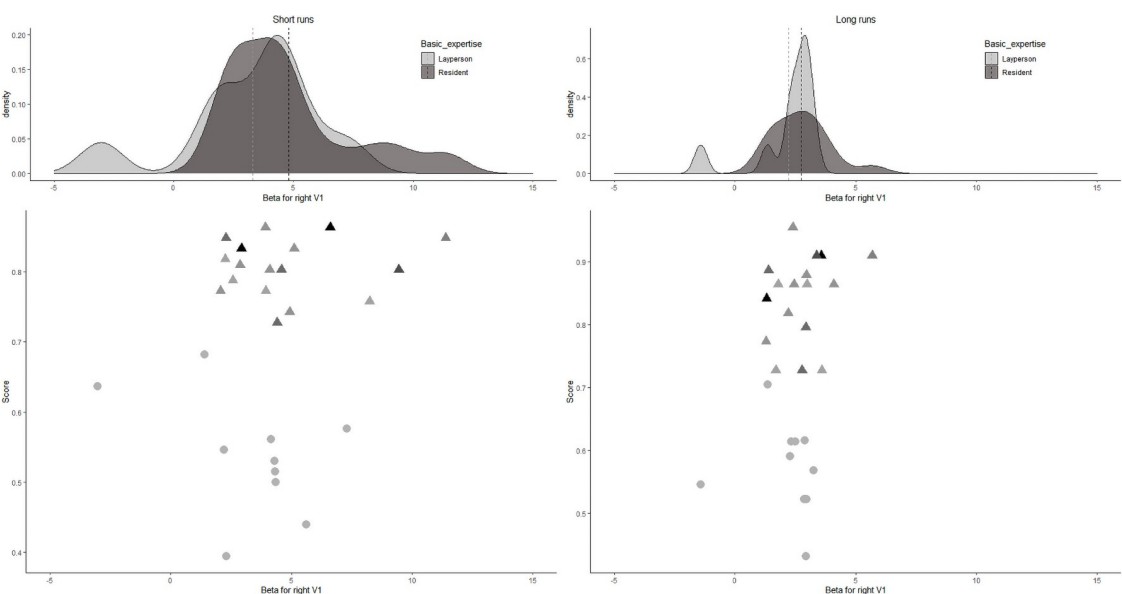

**Fig 7. Density plots (upper panels) and scatter plots (lower panels) for the individual beta values in the right V1 for short- and long-presentation trials, for correctly diagnosed trials only.** In the upper panels, the dotted line denotes the group mean. In the lower panels, for residents, darker colors depict more experienced residents.

We also added exploratory correlation analyses for the left FFA. There was a trend towards asignificant correlation between the beta values for short-presentation trials and the diagnostic performance for short-presentation trials in the left FFA ($r$ = -0.441, $p$ = .08), but not between the beta values for long-presentation trials and the average diagnostic performance for long-presentation trials ($r$ = -.263, $p$ = .31).

## Hypothesis 4: The activation level within the right FFA is (positively) correlated with experience level for radiologists in training

Spearman correlations between extracted beta values and the radiological experience of participants were calculated to take into account the ordinal nature of the experience measure. Within the radiologists-in-training group, no correlation between radiological experience and beta values were found for the short- ($Rho$ = 0.18, $p$ = .49) and the long-presentation trials ($Rho$ = 0.20, $p$ = .43).

We also added exploratory correlation analyses for the left FFA. There was no significant correlation between radiological experience and beta values in the short-presentation ($Rho$ = -.257, $p$ = .32) and the long-presentation trials ($Rho$ = -.378, $p$ = .13).

**Table 2. *F*- and *p*-values for the ANCOVAs.**

|  | Covariate | | Main effect expertise | |
| --- | --- | --- | --- | --- |
|  | *F* | *p* | *F* | *p* |
| short-presentation, all trials | 1.506 | .23 | 1.376 | .18 |
| long-presentation, all trials | .935 | .34 | .158 | .70 |
| short-presentation, only correct trials | 0.834 | .37 | 2.406 | .13 |
| long-presentation, only correct trials | .401 | .53 | .401 | .53 |

## Discussion

In the current study, we investigated the function of the right FFA in visual expertise tasks in radiologists in training. First of all, it was hypothesized that radiologists in training show higher diagnostic performance than laypeople. Second, it was hypothesized that the right FFA shows more activation for radiologists in training versus laypeople, in particular in the holistic mode (*i.e.*, during short-presentation trials) and less so in the search-to-find mode (*i.e.*, during long-presentation trials). Finally, it was expected that the activation of the right FFA is correlated with diagnostic performance and experience for radiologists in training for both modes.

In accordance with hypothesis 1, the diagnostic performance of radiologists in training was significantly higher than the diagnostic performance of laypeople for short- and long-presentation trials. We found tentative support for hypothesis 2, in the form of a significant three-way interaction between *expertise level*, *trial length*, and *location*. Radiologists in training were found to show somewhat higher involvement of the right FFA in diagnosing radiographs as compared to laypeople, during short-presentation trials but not during long-presentation trials, whereas the opposite pattern was found for the left FFA. However, none of the post-hoc t-tests showed significant differences between laypeople and radiologists in training. Additionally, there was a significant difference in the right FFA between laypeople and radiologists in training for short-duration trials when only correctly-diagnosed trials were included. In contrast to hypothesis 3, diagnostic performance did not correlate significantly with the beta values, and in contrast to hypothesis 4, participants' experience did not correlate significantly with beta values.

The analyses of V1 activity provide an insight into the attention account of expertise effect, which holds that expertise effects in the FFA and other brain regions are simply the effect of greater attentional engagement with the objects of expertise [9,18]. That is, an overall larger level of attention to objects of expertise by experts (because they are more interested in those stimuli) causes expertise effects not just in the FFA but also in other regions of the visual system, such as V1. Gauthier [35] argues against this account, showing expertise effects in the FFA even with limited attention (*e.g.*, when the object of expertise is irrelevant), and showing expertise effects even in the regional grey matter thickness of the FFA. To explore the attention account of expertise for our data, we investigated whether radiologists in training show larger V1 activation (as a result of larger attentional engagement) than novices and found no evidence for this. Additionally, as suggested by reviewers, we investigated whether partialling out V1 activation removes expertise effects, and found indeed that differences between conditions were no longer significant after partialling out V1 activity. This means that we cannot exclude that increased attention for the stimuli by residents compared to laypeople might explain the pattern of results in the FFA. On the other hand, the V1 activity is not a significant covariate in any of the ANCOVAs.

Most expertise studies concentrate on experts (*e.g.*, radiologists with at least ten years of experience) in visual tasks. In comparison, we found tentative evidence that the right FFA already responds to expertise-related stimuli in radiologists in training with only 1–5 years of experience in radiology, although the difference did not reach significance in all ANOVAs. This result suggests that fast holistic processes might play a role in diagnostic reasoning earlier than expected, as early as residency training. Of course, it has to be noted that our task was adapted to the level of the radiologists in training. We used abnormalities that were relatively easy to diagnose. Furthermore, we chose to present radiographs for two seconds, whereas experienced radiologists have been found to detect tumors on mammograms in as short as 250ms [1]. Still, these results suggest the involvement of the FFA in holistic processing in radiologists in training. While our design focused on another characteristic of holistic processing,

speed instead of the inversion effect, our results corroborate Bilalić's findings that the right FFA plays a crucial role in holistic processing.

These results provide further evidence for the idea that radiological expertise is reflected in the right-FFA activation level. Results, so far, are not completely consistent. Harley and colleagues [2] found no difference in right FFA-activation levels between expertise groups (1st-year residents versus 4th-year residents and practicing radiologists), but they did find a correlation between right FFA activation and diagnostic performance. We found the opposite pattern of results: we found differences in right FFA-activation levels between expertise groups (laypeople versus residents in training) but no correlation between right FFA activation and diagnostic performance or experience. This might be explained by the fact that we included a group of laypeople and a group of radiologists in training, whereas Harley and colleagues included no laypeople in their sample, but residents with two different expertise levels as well as a group of practicing thorax radiologists. Together, these results suggest that the right FFA starts responding to domain-specific stimuli already early in the process of acquiring expertise. Our lack of correlation between diagnostic performance and right FFA activation might be caused by the relatively high diagnostic performance of our radiologists in training. Participants were only required to indicate whether the potential diagnosis matched the diagnosis that they had in mind for the radiograph ('forced-choice' situation), which can be considered easier than diagnosing the abnormality. We instructed participants to execute their diagnostic reasoning during the presentation of the radiograph, and not during the presentation of the potential diagnosis. While this choice was made to optimize when the diagnostic process took place (*i.e.*, ensure that diagnostic reasoning took place during the presentation of the radiographs), the resulting diagnostic performance measure was suboptimal because it resulted in very high performance and this might explain our lack of correlation.

Findings in the literature are somewhat mixed when it comes to differences in right FFA activation between expertise groups in radiology. Neither Haller and Radue [30] nor Melo and colleagues [31] found areas with significant activation in the vicinity of the FFA. Note that they did not employ an FFA localizer task but only report areas with significant activation. Bilalić and colleagues [23] found this only to some extent. However, two of those studies [23,30] employed tasks that were not tapping into the process of diagnosing radiographs, but instead employed unrelated tasks that required participants to only look at radiographs. Harley and colleagues [2] required participants to detect the presence or absence of cued tumors, and only Melo and colleagues [31] required participants to formulate a diagnosis (but for shortly presented radiographs only). Diagnostic reasoning relies strongly on (structured) expert knowledge and is thus likely to result in different patterns of brain activation than tasks that can also easily be conducted by novices, such as a 1-back task [23] or the task to spot manipulations in radiographs [30]. The role of the right FFA in visual expertise might depend on which (expertise-related) task is executed, how much time was spent, and other processes such as top-down attention modulation [18].

Related to this is an important limitation of the study: holistic processing is a term that is notorious for its many definitions and associations [32, 36]. A similar concern is true for diagnostic processing, a term that includes many different cognitive tasks and processes [25, 31]. The specific design of the task and the instructions are thus very likely to impact what diagnostic processes take place, and this, in turn, impacts results. For example, while our task aimed to separate holistic processing from the search-to-find mode, it is very likely that some holistic processing has taken place in the long-presentation trials as well, in which participants were instructed to rely mostly on the search-to-find mode. In contrast, it is unlikely that the search-to-find mode can be executed in the short time of the short-presentation trials, so these trials are likely to reflect mostly holistic processing. Still, our aim of tapping into the complexity of

diagnostic processes made it relatively difficult to fully separate these two modes. Other researchers have avoided this problem by employing tasks with low similarity to the actual image interpretation task, such as 1-back tasks or detecting image manipulations. However, we consider it critical to go beyond this type of tasks and complement the literature by incorporating tasks that realistically reflect the diagnostic-reasoning process. Together, all these designs explore the complexity of the diagnostic-reasoning process. Thus, further research should use different types of tasks (including diagnostic-reasoning tasks) and designs to illuminate under which conditions results converge and diverge [*cf.* 37], in order to understand how visual expertise is represented in the right FFA. Likewise, the holistic processing of stimuli is only one aspect of visual expertise, and it would be interesting to also investigate the involvement of other regions of the brain in other aspects of visual expertise, such as areas in the occipital and frontal cortex, see, *e.g.*, [21] for example in radiology [2,30,31].

Another limitation of our study related to this aim is that participants were required to check their diagnosis based on the short-presentation runs during the long-presentation runs to elicit the checking mode. Thus, the order of short- and long-presentation runs could not be counterbalanced and we could not use a new set of images in the long-presentation runs. This could have caused order effects and/or effects of novelty. That is, all pictures in the long-presentation runs were already inspected in the short-presentation runs, which could have caused disengagement. Not only was the repetition of stimuli necessary to elicit the checking mode, by using the same stimuli in the long-presentation runs and short-presentation runs, we also ensured that the difficulty of the two modes was the same. Furthermore, it has to be noted that although the images were presented twice, even a presentation time of 10s is shorter than what residents would normally spend on an image, making disengagement an unlikely explanation of the lower activity during the longer trials. Anecdotally, participants remarked that they would have preferred seeing the images even longer. Finally, it has to be noted that we do not interpret the main effect of *trial length*, but only the main effect of *expertise* and the interaction of *expertise* and *trial length*. Even so, further research could counterbalance the order of the short-presentation and long-presentation runs, to entirely exclude a possible order effect. This would also require equally difficult stimuli in the two modes and accordingly adapted instructions in the long-presentation runs.

A final important limitation of our study is the limited number of participants. As a reviewer pointed out, this could also explain that we found the opposite pattern of results as Harley and colleagues [2]. Recently, it has been argued that those earlier studies on the expertise account of the FFA with small sample sizes have overestimated the expertise effects [15]. The issue of power in fMRI has received increasing attention [38], with suggestions to execute power analyses before data collection in a pilot study. This is still seldomly done and complex in expertise studies where eligible participant populations are small. In our study, two issues increased power: The blocked design and the sufficiently high number of trials. Unfortunately, our sample size, loosely based on those earlier findings, might have been too small given our current understanding of the size of the expertise effect. With this, our study gives tentative evidence that indeed expertise effects can be found in the right FFA in the holistic mode but less so in the search-to-find mode, but further research with larger samples is necessary to corroborate our findings.

In conclusion, our data provides tentative support for the general expertise hypothesis of right FFA functioning in a group of radiologists in training. On top of that, we found tentative support for the hypothesis that the right FFA shows more activation for radiologists in training versus laypeople, in particular in the holistic mode (*i.e.*, during short-presentation trials), and less so in the search-to-find mode (*i.e.*, during long-presentation trials). We did not find significant correlations between diagnostic performance and right FFA activation. These data

provide some evidence for the view that the right FFA supports holistic processing of stimuli in participants' expertise domain.

## Acknowledgments

We would like to thank Armin Heinecke from BrainInnovation B.V. (Maastricht, the Netherlands) for his help with the *BrainVoyager* analyses and Sven Gijsen for programming the visual stimulation using the *BrainStim* software.

## Author Contributions

**Conceptualization:** Ellen M. Kok, Andreas Gegenfurtner, Jeroen J. G. van Merriënboer, Anique B. H. de Bruin.

**Data curation:** Ellen M. Kok, Bettina Sorger.

**Formal analysis:** Ellen M. Kok, Bettina Sorger.

**Funding acquisition:** Ellen M. Kok, Jeroen J. G. van Merriënboer, Anique B. H. de Bruin.

**Investigation:** Ellen M. Kok, Bettina Sorger, Koos van Geel, Andreas Gegenfurtner, Anique B. H. de Bruin.

**Methodology:** Ellen M. Kok, Bettina Sorger, Andreas Gegenfurtner, Jeroen J. G. van Merriënboer, Simon G. F. Robben, Anique B. H. de Bruin.

**Project administration:** Ellen M. Kok.

**Resources:** Simon G. F. Robben.

**Supervision:** Bettina Sorger, Jeroen J. G. van Merriënboer, Simon G. F. Robben, Anique B. H. de Bruin.

**Validation:** Bettina Sorger.

**Visualization:** Ellen M. Kok.

**Writing – original draft:** Ellen M. Kok.

**Writing – review & editing:** Bettina Sorger, Koos van Geel, Andreas Gegenfurtner, Jeroen J. G. van Merriënboer, Simon G. F. Robben, Anique B. H. de Bruin.

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
