## [Decision Letter · Decision Letter 0]

7 Sep 2020

PONE-D-20-23315

Holistic processing only? The role of the fusiform face area in radiological expertise

PLOS ONE

Dear Dr. Kok,

Thank you for submitting your manuscript to PLOS ONE. After careful consideration, we feel that it has merit but does not fully meet PLOS ONE’s publication criteria as it currently stands. Therefore, we invite you to submit a revised version of the manuscript that addresses the points raised during the review process.

There are a fair number of comments to be addressed by the Reviewers but they all seem highly pertinent to improve the overall quality of the manuscript. In addition, please carefully proofread the manuscript as there are some typos throughout (e.g. "Them," instead of "Then," in the abstract). Also, I recognize that recruiting additional subjects for the study, as suggested, may not be feasible. However, please do your best to discuss this aspect within your manuscript. Please also be careful about the use of the term "marginally significant". While you can discuss these results, the use of the term "trend" is more appropriate (e.g. lines 319-320).

We look forward to receiving your revised manuscript.

Kind regards,

Niels Bergsland

Academic Editor

PLOS ONE

Journal Requirements:

2.We note that you have indicated that data from this study are available upon request. PLOS only allows data to be available upon request if there are legal or ethical restrictions on sharing data publicly. For more information on unacceptable data access restrictions, please see http://journals.plos.org/plosone/s/data-availability#loc-unacceptable-data-access-restrictions.

Reviewers' comments:

Reviewer's Responses to Questions

**Comments to the Author**

1. Is the manuscript technically sound, and do the data support the conclusions?

Reviewer #1: Partly

Reviewer #2: Yes

2. Has the statistical analysis been performed appropriately and rigorously? 

Reviewer #1: Yes

Reviewer #2: Yes

3. Have the authors made all data underlying the findings in their manuscript fully available?

Reviewer #1: No

Reviewer #2: Yes

4. Is the manuscript presented in an intelligible fashion and written in standard English?

Reviewer #1: Yes

Reviewer #2: Yes

5. Review Comments to the Author

Reviewer #1: The authors test whether the right FFA exhibits experience related responses to radiographs in radiology students vs a set of controls. The authors replicate the expertise-rFFA link like many papers before it, but only in the blocks where participants have to respond quickly. They do not find similar expertise related differences in V1, which the authors claim rejects the attention hypothesis of FFA expertise effects.

While it’s always reassuring to see a finding in the literature replicated, I think the authors could extend the literature in a more novel way by localising and examining these effects in other brain regions. I also have some comments on the authors’ interpretations of their data, analysis and methodology. The study is low powered and the authors should acknowledge this or remedy it with more data (although I appreciate this latter solution may be impossible due to lack of funding). I should mention I am not an expert on the technical minutia of fMRI data recording or processing. This paper should therefore be reviewed by an fMRI expert too, one who can assess the technical aspects carefully. However, I will comment on these as best I can. Having read all fMRI expertise papers, I am much more familiar with the paradigms used, and the evidence that supports and rejects the expertise hypothesis, which I feel will help the authors in my review.

Major points

1. While replicating non-face FFA effects is interesting, it has now been shown in around 20-30 papers. One thing that is less clear is whether the left FFA or either OFAs are responsive to object expertise. When you examine the literature, it is only a tiny minority of papers that show effects in these regions (e.g., Harley et al., 2009; McGugin, Newton, Gore, & Gauthier, 2014; McGugin, Van Gulick, Tamber-Rosenau, Ross, & Gauthier, 2014; Ross et al., 2018). Why do the authors not try and remedy this by localising these other regions and exploring whether they are also responsive to expertise? Or, if they can’t, the authors should report why; i.e., maybe they can’t localise these regions, which has been mentioned in some papers. The same is also true of effects in the LOC too.

I note the authors mention that they only looked at the FFA to avoid employing multiple comparisons with other areas that would inflate the risk of a Type 2 error, but this is not a good reason to not perform these additional analyses. Burns, Arnold, & Bukach (2019, Footnote 4) found that conservatively speaking, expertise studies should test 56 participants even when engaging one-tailed analyses. That would place this study as underpowered with n = 28, or 17 experts (I should mention only one expertise study does not suffer this low power problem: Martens et al., 2018). It does not therefore make sense to perform corrections for multiple comparisons when power is low (Nakagawa, 2004). Instead, it would be more interesting to run the analyses on the other localised regions, and simply report that due to low power, the p-values will not be corrected. That way, we can see if there are expertise effects in other regions (which are not often reported in the literature outside of the rFFA). Also, were the reported analyses two-tailed?

2. Relatedly, how did the authors determine sample sizes? Did they analyse their data as each participant was recruited then stop when they found predicted results? Or was there an a priori number? It seems the sample size is similar to that suggested in the post 2010 Gauthier lab papers, but the authors do not specify this.

3. The authors say there is a need for a non-passive study, but this doesn’t sound right. When I double checked, citation 2 (Harley et al) already did a diagnostic fMRI task in radiologists. While their localizer scan was passive, which is common, in the actual experiment the participants tried to identify the nodules (from the paper follows):

“The diagnosis scans were event-related scans in which participants viewed intact and coarsely scrambled radiographs and judged whether a cued region in each radiograph contained a lung nodule”

This means what the authors say has not been done, actually has been. The more novel aspect of their study therefore is the 2s vs 10s difference. I think the authors may want to reread their other citations to make sure there aren’t other errors when describing the literature.

4. I can’t think of a single fMRI expertise study that has made their raw data publically available even though there are repositories available for this very purpose (D’Esposito, 2000). Why do the authors not make their data available? It would help start a precedent, improve transparency and replicability, and future researchers could reuse the data with other sets to increase power. Or do the authors have a particular reason for not sharing their data?

5. Regarding the correlations between expertise and neural activity, I think it would help if they were actually plotted (they could go into sup matts if the authors prefer). That way the reader could see individual data points. Also, a table of the within group and across groups correlation co-efficients and p-values within each condition/location would also be helpful. Gauthier and colleagues in their most recent papers recruit participants with a broad range of expertise (i.e., they’re not all experts) and find correlations between activity and FFA activity, so having all participants analysed together would help increase power.

6. It is common for researchers to find the voxel in the FFA that activates the most to faces, and then test whether this is also the peak for expertise, did the authors do this here? I can’t see mention of it.

Minor points

Page 3 line 60, I’m not sure there is a general consensus. I still meet countless people at conferences/talks who claim the expertise hypothesis is not real. This is reflected in the literature where Kanwisher (2017) still argues against these effects, with echoes of this present in another recent review (Duchaine & Yovel, 2015).

The authors touch upon prosopagnosia, so it may be worth noting that there are three separate studies from Jason Barton’s lab of groups of patients with FFA lesions that exhibit impaired object expertise (the citations are in the recent Burns et al 2019 expertise meta-analysis paper). One of these papers doesn’t specify the patients have FFA lesions, but if you check their earlier papers, you can identify that the cases actually do. I think when combined with the fMRI data, the expertise hypothesis becomes very compelling. In the intro the authors ask what the FFA is doing in expertise, and I think this data strongly suggests it contributes to the experience based individuation of highly similar object exemplars.

The authors refer to chest pieces, but I think they mean chess.

Page 4

The authors don’t really define holistic processing (or many other terms they introduce). What is it? Does inversion disrupt it because we are no longer using experience based pathways which would typically recognise such objects in its commonly seen upright configuration? Where’s the evidence that speed is a more central feature of holistic perception than stimulus configuration? What is search to find? What are the conceptual similarities pointed out by Bilalic? It’s not explained clearly why short versus long duration should differentiate experts in their neural activity?

Why is the FFA the most theoretically relevant? How do we know it’s the most relevant if the other regions (e.g., OFA, lFFA) have not been not as well studied?

Page 14

I presume the authors look at V1 because others have suggested FFA expertise activity is an artefact of attention, however, the authors do not explain this, nor cite the previous literature positing this (Harel et al, 2010; Kanwisher, 2017). Prior work has tried to address this issue (McGugin et al., 2015; McGugin et al., 2016) and there are other problems with this hypothesis (Burns et al., 2019).

Change level should be chance level?

Page 16

Marginally significant effect was not significant and so should be reported as such (p = .052). Nor is the later interaction significant (p = .08). Despite this latter result, I’m guessing the subsidiary analyses were preplanned based upon the authors’ hypotheses and prior literature, hence the motivation to perform them despite the interaction not being significant? If so, this should be explained.

When performing the correlations between FFA activity and performance, would it not be interesting to partial out the V1 activation? If the authors argue this activity is attention related, then taking it into account may yield a non-attention related relationship between FFA and expertise? The authors could note these were exploratory as they were not planned. The logic behind this type of analysis is detailed in DeGutis, Wilmer, Mercado & Cohen (2013) and has been used in many of the McGugin FFA expertise papers. While they regressed out different types of behavioural performance, and I haven't seen it used to regress out brain activity, I believe it the same logic could be applied and be useful here.

The correlation coefficients of around .2 are actually at the most conservative end of the predicted expertise-FFA relationships when the broad literature is taken into account (Burns et al., 2019; Footnote 4). Again hinting the effect may be there, it’s just that insufficient numbers were tested. As mentioned earlier, why not run the correlations across groups (and plotted), as it would increase power and give a broader range of expertise.

I find it strange the authors did not compute dprime as this was what was correlated with the FFA in the Harley paper.

Page 20

A simpler explanation for the different pattern of significant results in this and the Harley paper is that both were underpowered (Burns et al., 2019). If both studies only have 50% power and run two different analyses testing for a real effect, then it’s not surprising if only one of them is significant and the other not. If power was extremely high (>99%), we would expect both to almost always be significant. This in my mind is the most likely explanation for why analyses on the link between FFA and expertise can be inconsistent even within a paper, with authors giving quite complicated explanations for incongruent results (and this occurs in almost all expertise papers). Instead, low power is the simplest and most likely explanation.

Page 21

The Haller and Melo papers did not test the FFA (i.e., no localiser task) so it is not accurate to report this as such.

Reviewer #2: The authors described their work on an interesting question that whether FFA is responsible for only holistic processing of expertise images but not for slower processing mode, which is a new angle lacks research. They have used a new set of task that required the subjects to do a diagnose based on a picture, which is better representing the real-life processing of the images, which engages attention, serial search and other reasoning.

The results seemed solid, only with big variations between subjects. Error bars are huge. Recruiting more subjects, getting rid of the subjects with big movement artifact would largely improve that. The correlation analysis may also improve if more subjects included. It’s also good if they can show individual beta values in figure 4.

There is one issue with the design though. As is understandable that it is always more difficult to claim a negative conclusion. The authors found FFA response in expert group is lower during 10s task than 2s task. They contributed such difference into FFA’s selectivity to short presentation. However, such difference could also due to: 1. Novelty effect/task difficulty. All pictures in 10s task were already seen in 2s task. Therefore less engaged, worse attention. 2. Longer exposure of the same image adapted the visual system, and BOLD signal became weaker. Such general mechanisms could contribute to the difference. And the V1 response also showed similar trend – 10s trials showed lower response than 2s trials, which again indicated such difference is not a unique nature of FFA processing, but could be contributed to the task or BOLD signatures.

If the authors could make the 10s task more challenging, I doubt FFA may have bigger response even with a slow processing mode. For example, if the 10s task use new set of images (not seen ones), and the subjects need to find the abnormalities as many as possible (not a yes or no question), or it's a new set of more difficult images (rather than easy typical educational examples), the subjects may engage more in the task.

The author didn’t mentioned whether they tracked the eye position or not, nor did they mention whether the subjects are required to fixate or free viewing. I assume that the subjects will make more saccades during the 10s task, and that may cause more variations in to the overall signal.

Other typos.

Line 68, 69: chess, not chest

Line 315: Figure 4 (right)

6. PLOS authors have the option to publish the peer review history of their article (what does this mean?). If published, this will include your full peer review and any attached files.

Reviewer #1: No

Reviewer #2: **Yes: **Haoran Xu

---

## [Author Response · Author response to Decision Letter 0]

5 Nov 2020

Dear editor,

Please find the response to reviewer and editor comments in the response to reviewers letter.

Kind regards, Ellen Kok

---

## [Decision Letter · Decision Letter 1]

22 Dec 2020

PONE-D-20-23315R1

Holistic processing only? The role of the fusiform face area in radiological expertise

PLOS ONE

Dear Dr. Kok,

Thank you for submitting your manuscript to PLOS ONE. After careful consideration, we feel that it has merit but does not fully meet PLOS ONE’s publication criteria as it currently stands. Therefore, we invite you to submit a revised version of the manuscript that addresses the points raised during the review process.

Upon review of the revised version, the Reviewers and myself continue to have concerns about the current manuscript. I agree with both Reviewers that the V1 results should not have been removed from the revised manuscript. Please pay careful attention to respond to each of the points that have been raised. Although there is a non-negligible amount of work to do, I trust that the detailed reviews and feedback from the Reviewers will help facilitate your revision.

We look forward to receiving your revised manuscript.

Kind regards,

Niels Bergsland

Academic Editor

PLOS ONE

Reviewers' comments:

Reviewer's Responses to Questions

**Comments to the Author**

1. If the authors have adequately addressed your comments raised in a previous round of review and you feel that this manuscript is now acceptable for publication, you may indicate that here to bypass the “Comments to the Author” section, enter your conflict of interest statement in the “Confidential to Editor” section, and submit your "Accept" recommendation.

Reviewer #1: (No Response)

Reviewer #2: (No Response)

2. Is the manuscript technically sound, and do the data support the conclusions?

Reviewer #1: Partly

Reviewer #2: Partly

3. Has the statistical analysis been performed appropriately and rigorously? 

Reviewer #1: No

Reviewer #2: Yes

4. Have the authors made all data underlying the findings in their manuscript fully available?

Reviewer #1: No

Reviewer #2: Yes

5. Is the manuscript presented in an intelligible fashion and written in standard English?

Reviewer #1: Yes

Reviewer #2: Yes

6. Review Comments to the Author

Reviewer #1: The authors have edited their manuscript to correct some issues, but some of their responses need further clarification, while other haven’t addressed my original comments. I realise my major points are quite long this time, but this is just because I want to ensure I’m clear enough so that we avoid any misinterpretations. I hope the authors don’t think I’m being intentionally difficult or disingenuous, it certainly isn’t my intention. I just want to ensure the decisions the authors have made in their methods and analyses are transparent to future readers, and that there is not a risk of readers misinterpreting how the literature currently stands with respect to power considerations.

Major points

1. The authors state that they can’t look at the lFFA or OFAs because they never intended to study them in the first place. Yet they have now removed a priori research questions. See the authors’ two responses to reviewers:

“Although your suggestion to localize and analyze other regions is certainly interesting, it was not the original goal of the project”

“in line with their argumentation, we have decided to remove the V1 analyses from the paper completely.”

I agree it’s perfectly fine to focus your manuscript on what the original research questions were, and these should be reported as a priori hypotheses. What is problematic is that these claims are at odds with the authors removing their original V1 hypotheses and analyses after the first review. These have completely vanished, as if they never existed. Yet this was exactly what the authors planned. The replication crisis has shown us that removing or altering hypotheses post-hoc to fit the data promotes false positives (seems a form of HARKing, Kerr, 1998, where authors drop a priori hypotheses). At the most fundamental level, the V1 data must be reinstated in the manuscript.

Following this train of thought, why can’t they at least attempt analyses on the lFFA, and/or the OFAs, and report that these as exploratory (or report an inability to localise). You’ve spent a lot of time, effort and expense acquiring this data, so why not report it (it would also add to the novelty, provide effect sizes or illustrate localisation issues which are common in the literature for other researchers)? Especially when the raw data are not immediately available for others to answer these questions themselves.

I’m not trying to be difficult, but these competing arguments over original research questions and post-hoc questions do not accord with one another.

2. I don’t think it will affect the importance of the manuscript if the V1 data is linked to the FFA effect, it is better to be transparent so readers can make up their own minds. This is why performing correlations between V1 and FFA activity are interesting (same is true with the partial out method). As I mentioned previously, even if there is a link between V1 and FFA effects, this is still consistent with the expertise hypothesis: see Gauthier’s recent commentary on Kanwisher’s paper on the history of FFA research, and Lohse et al. (2016) on this link to face regions. If the face processing link is expertise based, it makes sense if there’s a link due to non-face expertise. You should report these as exploratory.

3. Can’t the authors ‘deface’ the raw fMRI data so they remove the risk of identification? Doesn’t this mean they could then make the data publically available? Also, I can’t access the data file the authors provide to replicate their analyses.

4. The description in the Methods of how the authors decided their sample size still doesn’t appear to be based on any objective criteria. Saying you decided to test 10 in this group and 10 in another group is not the same as using power estimates from prior work to decide sample sizes. I am mindful that pushing this issue further may make a posthoc decision appear, but why were these numbers decided (the authors mention line numbers 168-170, but I don’t see this in my pdf, maybe they mean 147-149), was power ever considered when designing your study? The decision to stop at 17 experts for practical reasons is again too vague. What were these reasons? Funding spent elsewhere? Funding expiring? Scanner was no longer available to the researchers? You asked all your radiologists and the others said no? Or you stopped as you found a significant effect? It should be stated.

5. The following points all touch upon issues of power that the authors discuss.

The authors cite Gegenfurtner et al. in their discussion, which is a review of eye tracking effect sizes, but the current study is fMRI. I don’t think this is appropriate and should be removed as it reminds me of the Cow-Canary problem where effects can be massively different between two different methods (Capitani et al., 1999). The authors’ own behavioural data here shows a huge effect size (which would be expected between experts and novices), yet only a medium one in the fMRI data. Why discuss eye-tracking effect sizes when you had 20-30 fMRI expertise studies to base your decision on, it doesn’t make any sense in this context and should be removed as just confuses the issue.

In the same place in the Discussion the authors seem tacitly argue two fMRI expertise studies that are 20 years old are an acceptable guide for sample sizes (Gauthier et al., 1999: which they don’t cite here but describe, and Gauthier et al., 2000). This ignores the recent studies that used larger sample sizes (peaking with the recent Martens et al paper, which is not cited, nor most other studies in the last 10 years). Even if two 20 year old papers were the basis for making their sample size decisions, it should be mentioned in the Methods and not the Discussion (although again, a question remains as to why the authors selected such old papers to base their sample sizes on rather than more recent work).

From the above I hope it’s clear why the current Discussion paragraph on effect sizes from eye tracking and two 20 year old fMRI papers does not work. I don’t want future researchers reading this paper to think these sample sizes would lead to sufficient levels of power when designing their own fMRI expertise studies. I think it would be more helpful to explicitly remind the readers that we now know this study and Harley’s were likely underpowered, that classic papers from 20 years ago were underpowered, and that significant effects here were found using liberal alphas. It’s perfectly fine accepting the limitations of this manuscript, but they need explained to the readers so they’re aware of the issues.

6. Also related to the previous points. Maybe I was not clear enough in my first review, but when I referred to page 20 and the different patterns of results, I meant that you failed to find a correlation but Harley and colleagues did. However, you found a group experience difference that Harley didn’t. The simplest explanation for this disparity is that both studies are underpowered. If you have two studies that are underpowered (maybe 50ish%), then it’s likely one (Harley et al) will find a significant effect in one analysis, but not the other, while the second paper (the current manuscript) finds the opposite pattern of significant/non-significant results with the same analyses. You haven’t discussed this in this paragraph, nor in the later paragraph where you argue your study is not underpowered based on two studies from 20 years ago. You should explain this hypothesis of the conflicting results explicitly when discussing the differences between Harley and yourselves.

I should add, the discussion of the 2s vs 10s trials is distinct from the point I’m trying to make here (although a similar logic could be applied, I think the correlation/group difference disparity needs explained separately).

Minor points

1. The authors uploaded two different versions of their manuscript, one with tracked changes indicated and another without, but switched between them when they were referring in their response to reviewers (and they sometimes referred to line numbers that didn’t correspond with the text). This made checking between the different texts confusing and overly time consuming to review (and apologies if I’ve now referred to one text and the other). I’d recommend on the next submission they simply change the font color of the text they’d edited and upload a single manuscript (unless this is a journal specific requirement to upload two?). Same is true of having the figure legends next to the actual figures.

2. I think the authors have misinterpreted signal detection theory by claiming they can’t compute dprime. They employed a 2AFC task, with stimulus pairs that were same or different. This leads to categories of hits, misses, CRs and FAs. It’s quite easy to compute dprime to test whether this behavioural measure is more strongly correlated to FFA activity as it was in Harley et al. I’m convinced other readers would want to see this to compare the results between the current study and Harley et al.

3. Line 84: “processing a face as a whole, and not as a combination of features”

I think referring to ‘combination of features’ implies a holistic relational percept. Maybe something like “as separate, distinct features that do not interact to form a single percept” is clearer?

4. Removing the original graphs stops making the group differences/similarities in the different conditions easily comparable so should be reinstated. I like the individual plots, but trendlines would be a useful addition too.

5. Line 490 still reads like prior studies analysed an area that included the FFA (i.e., you mention overlap), but we can’t say this as they did not localise it. It could be in the vicinity of the FFA, but not actually overlap . Related to this, there are many expertise FFA studies that did localise the FFA and are not cited in this manuscript which seem more relevant (particularly the Martens et al study which should be the gold standard going forward). Surely at least from a point of generating interest in the current paper the authors would want to reference these as more citations lead to more exposure?

Reviewer #2: I appreciate the authors modified and responded to each of the points I raised in the review.

My remaining concern for the paper is obviously the insignificance in the difference of rFFA activation between experts and laypeople for 2s trials. The statistics showed that for all trials, Ttest p=0.06, for correct trials ttest p=0.04, ANOVA results for expertise vs laypeople are all insignificant. Such small effect could be caused by the two dots shown in fig 4 and 5, which showed around 2 and 3 in beta values, but their diagnostic performance were low, and the training background were short. Given all the main conclusions and novelties for this paper are based on this small “trend”, and there seems no way the authors could recruit more subjects and alter that, I therefore am not quite confident to believe what have been concluded in the paper.

Reviewer #1 pointed out a paper by Burns, Arnold and Bukach 2019 in Neuroscience and biobehavioral reviews, in which the authors tried to justify p-values close to 0.05 could still be significant by meta-analysis. I am however not convinced that such meta-analysis on different tasks could justify the current small “trend” seen in the paper under review.

I also don’t quite like the author removed the V1 results. Like reviewer #1 suggested, I think the authors should use V1 to normalize the rFFA response, to control the possible attention/engagement bias in both subject groups and expertise levels. Literatures have shown that early visual areas including LGN, V1 and V4 all show attentional modulations in fMRI studies.

7. PLOS authors have the option to publish the peer review history of their article (what does this mean?). If published, this will include your full peer review and any attached files.

Reviewer #1: No

Reviewer #2: No

---

## [Author Response · Author response to Decision Letter 1]

15 Apr 2021

Dear editor and reviewers,

Please find our responses to your comments in the file 'Response to reviewers'.

Kind regards, Ellen Kok

---

## [Decision Letter · Decision Letter 2]

21 Jun 2021

PONE-D-20-23315R2

Holistic processing only? The role of the fusiform face area in radiological expertise

PLOS ONE

Dear Dr. Kok,

Thank you for submitting your manuscript to PLOS ONE. After careful consideration, we feel that it has merit but does not fully meet PLOS ONE’s publication criteria as it currently stands. Therefore, we invite you to submit a revised version of the manuscript that addresses the points raised during the review process.

I would also like to thank you for your patience during the review process as I recognize it has been quite lengthy.

We look forward to receiving your revised manuscript.

Kind regards,

Niels Bergsland

Academic Editor

PLOS ONE

Journal Requirements:

Reviewers' comments:

Reviewer's Responses to Questions

**Comments to the Author**

1. If the authors have adequately addressed your comments raised in a previous round of review and you feel that this manuscript is now acceptable for publication, you may indicate that here to bypass the “Comments to the Author” section, enter your conflict of interest statement in the “Confidential to Editor” section, and submit your "Accept" recommendation.

Reviewer #1: All comments have been addressed

2. Is the manuscript technically sound, and do the data support the conclusions?

Reviewer #1: Partly

3. Has the statistical analysis been performed appropriately and rigorously? 

Reviewer #1: Yes

4. Have the authors made all data underlying the findings in their manuscript fully available?

Reviewer #1: Yes

5. Is the manuscript presented in an intelligible fashion and written in standard English?

Reviewer #1: Yes

6. Review Comments to the Author

Reviewer #1: I think the manuscript reads better now. I only have one main comment:

In the previous drafts, the authors ran a one-tailed t-test on the right FFA short run which showed an effect of expertise, but for some reason they are no longer reporting this as one-tailed, but two-tailed now on line 401 (I think, the t-value is identical and the p-value is now double what it was previously in the deleted section), which renders the effect of expertise non-significant. I can't work out why this has been changed? I think it would be beneficial if this was changed back to the one-tailed test as that was what the authors initially intended and corroborates what they're actually claiming, i.e., an effect of expertise in the right FFA. There may need to be some minor text adjustments because of this (e.g., paragraph at line 466).

Lines 481-483 need citation/s.

There were a couple minor typos so a proofread may be good (e.g., line 77 gap before comma).

7. PLOS authors have the option to publish the peer review history of their article (what does this mean?). If published, this will include your full peer review and any attached files.

Reviewer #1: No

---

## [Author Response · Author response to Decision Letter 2]

28 Jul 2021

Dear editor,

Thank you for providing us with the opportunity to revise the manuscript. Below, we detail how we have adapted the manuscript in response to the comments of the reviewers. Additionally, the reference list was checked for completeness and correctness.

Kind regards on behalf of all authors,

Ellen Kok

1. In the previous drafts, the authors ran a one-tailed t-test on the right FFA short run which showed an effect of expertise, but for some reason they are no longer reporting this as one-tailed, but two-tailed now on line 401 (I think, the t-value is identical and the p-value is now double what it was previously in the deleted section), which renders the effect of expertise non-significant. I can't work out why this has been changed? I think it would be beneficial if this was changed back to the one-tailed test as that was what the authors initially intended and corroborates what they're actually claiming, i.e., an effect of expertise in the right FFA. There may need to be some minor text adjustments because of this (e.g., paragraph at line 466).

We had adapted the analysis to a two-sided test because all other tests were two-sided. Since we originally had a one-sided test (because we had a specific hypothesis, we have now changed the sentence to read:

“Post-hoc t-tests show a significant effect for the right FFA short runs, t(26) = 1.80, p = .04 (one-sided t-test), Cohen’s d = 0.71 (medium-to-large effect); for the other t-tests all t’s <1.0.” (lines 401-402 in the version without tracked changes).

Additionally, we changed ‘trend towards significance for the difference’ to ‘significant difference’ in line 471 in the version without tracked changes.

2. Lines 481-483 need citation/s.

We have now added the citation that was accidentally left out for line 481-482 (reference 35).

3. There were a couple minor typos so a proofread may be good (e.g., line 77 gap before comma).

An additional thorough proofread was conducted, fixing the typo in line 77 and several others.

---

## [Editor Report · Decision Letter 3]

18 Aug 2021

Holistic processing only? The role of the right fusiform face area in radiological expertise

PONE-D-20-23315R3

Dear Dr. Kok,

We’re pleased to inform you that your manuscript has been judged scientifically suitable for publication and will be formally accepted for publication once it meets all outstanding technical requirements.

Kind regards,

Niels Bergsland

Academic Editor

PLOS ONE
---

## [Editor Report · Acceptance letter]

23 Aug 2021

PONE-D-20-23315R3 

Holistic processing only? The role of the right fusiform face area in radiological expertise 

Dear Dr. Kok:

I'm pleased to inform you that your manuscript has been deemed suitable for publication in PLOS ONE. Congratulations! Your manuscript is now with our production department. 

Kind regards, 

on behalf of

Dr. Niels Bergsland 

Academic Editor

PLOS ONE